# Analysis and Visualization of Vessels' RElative MOtion (REMO)

Hyowon Ban [1] and Hye-jin Kim [2,*]

1   Department of Geography, California State University Long Beach, Long Beach, CA 90840, USA
2   Korea Research Institute of Ships & Ocean Engineering, Daejeon 34103, Republic of Korea
*   Correspondence: hjk@kriso.re.kr; Tel.: +82-42-866-3114

**Abstract:** This research is a pilot study to develop a maritime traffic control system that supports the decision-making process of control officers, and to evaluate the usability of a prototype tool developed in this study. The study analyzed the movements of multiple vessels through automatic identification system (AIS) data using one of the existing methodologies in GIScience, the RElative MOtion (REMO) approach. The REMO approach in this study measured the relative speed, delta-speed, and the azimuth of each vessel per time unit. The study visualized the results on electronic navigational charts in the prototype tool developed, V-REMO. In addition, the study conducted a user evaluation to assess the user interface (UI) of V-REMO and to future enhance the usability. The general usability of V-REMO, the data visualization, and the readability of information in the UI were tested through in-depth interviews. The results of the user evaluation showed that the users needed changes in the size, position, colors, and transparency of the trajectory symbols in the digital chartmap view of V-REMO for better readability and easier manipulation. The users also indicated a need for multiple color schemes for the spatial data and more landmark information about the study area in the chartmap view.

**Keywords:** maritime transportation; expert evaluation; nautical charts; relative motion; situation awareness system; geographic information science; cartographic visualization

## 1. Introduction

Maritime transportation in port control areas can be complex when several vessels operate, anchor, or ship. Therefore, detecting unusual maritime traffic situations becomes easier for traffic controllers if they can observe the spatio-temporal characteristics of the transportation and operation of vessels—such as speed and azimuth—in terminal control areas.

It is necessary for vessel traffic service (VTS) operators to constantly be informed about the status of multiple vessels' operations and make decisions to prevent any possible maritime traffic accidents. However, it is challenging for traffic controllers to constantly observe large amounts of information that changes through space and time, so they may generate errors in their decision making about maritime traffic. Therefore, it is vital to develop a system that supports users—for example, traffic controllers—to recognize the characteristics of maritime traffic patterns and visualize marine traffic information from a large amount of data. The system can use information on the course over ground (COG) and speed over ground (SOG) of the vessels [1,2].

Recently, maritime transportation accidents have increased for various reasons. Some accidents have occurred because of overloading; technical problems; collisions with other ships; weather and water conditions; and social, economic, and political structures [3,4]. Examples include the Le Joola disaster in 2002, the Shariatpur-1 disaster in 2012, and the Sewol Ferry disaster in 2014 [4–7]. Especially in the case of South Korea, the number of maritime traffic accidents increased from 1093 cases (1306 vessels involved) to 2307 cases (2549 vessels involved) between 2013 and 2016 [8].

In the vicinity of ports, vessels are on course for entry and departure to and from the ports. Buoys only define the entry and departure courses, and vessels need to sail within

the extent of these courses. The direction of currents and the motion of seawater, such as tidal movements in port areas, influence the sailing of the vessels. Operators of vessels understand such characteristics of seawater, and they sail by manipulating the speed and azimuth of the vessels based on this knowledge. Additionally, traffic controllers monitor the operation of vessels in port areas, often controlling them by providing restrictions or advice [9].

Outside of ports, port control centers do not control vessels. Instead, they operate by themselves based on consideration of the seawater, seafloor geomorphology, and other vessels nearby. Therefore, it is necessary to manage information on traffic situations every time a vessel moves. Automatic identification system (AIS) data are helpful in analyzing marine traffic because they provide information on high temporal resolutions collected per second. AIS data consist of a vessel's position, identity, speed, course with other nearby ships, satellites, and so on. Usually, the resolution of the AIS data may range from low (sampling rate in minute unit) to high (sampling rate in second unit) [10]. The current study uses AIS data to deal with maritime transportation.

Recently, there has been much research regarding the risk analysis of maritime transportation, some of which has dealt with the methodology and applications of the analysis of maritime transportation [11]. For example, AIS data have been used to locate vessels by exchanging data with each other, including vessels' position, identity, speed, course with other nearby ships, AIS base stations, and satellites. Further, AIS data have been widely studied in relation to the topics of spatio-temporal distributions, anomaly detection, the prediction of the routes of vessels, the spatial domain of ships, the statistical analysis of traffic patterns and the collision risk of vessels, the emission estimation of ships, the uncertainty of the AIS data, and so on [12–18].

There exist decision support systems for maritime traffic. Many of them display the current locations of vessels on digital charts and generate an alarm sound when distances among the vessels decrease or signals regarding the locations of the vessels show abnormal patterns. Additionally, the systems show the locations of the vessels on charts. However, few provide more detailed information, such as analyses of each vessel's operations or maritime traffic in areas. Due to this, it is challenging for traffic controllers to recognize and control complex maritime traffic situations in the relevant areas [1,2].

The current research is a pilot study aiming to develop a maritime traffic control system that supports control officers' decision-making process and to conduct a usability test of the system. We analyze multiple vessels' movements using one of the existing approaches in GIScience, the RElative MOtion (REMO) approach, and visualize the results on two-dimensional, electronic navigational charts. Furthermore, we assess the tool via user interviews, which is the novelty of this study. Ultimately, the study aims to answer the following questions: (i) What are the crucial characteristics of a maritime traffic control system to support control officers in making decisions? (ii) How can multiple vessels' navigation information at a port be effectively analyzed and visualized for control officers? To answer these questions, we analyze AIS data and visualize the results using the REMO approach and GIScience following Laube and Imfeld [19]. The AIS data used in this study were collected from the Korea Research Institute of Ships and Ocean Engineering (KRISO).

*Background*

AIS data have been widely used to analyze and report vessels' movements. As a result, maritime traffic management has improved information processing, integration, and presentation using AIS data [1]. However, a large number of vessels' existing control systems rely on experienced officers' mental capacity for information analysis and decision making [20]. Additionally, the number of skilled workforces is limited, so officers can be prone to errors in decision making that may lead to marine traffic accidents [21]. Therefore, to increase the safety of marine traffic, it is necessary to develop enhanced control systems

that support the analysis of vessels' spatio-temporal distributions, and the prediction of routes and collision risk [12].

Several works have been conducted to aid the guidance and decision making of vessels. For instance, by using AIS data, the drifting of ships can largely influence marine traffic accidents [22]. Moreover, a reference model of port information systems was developed consisting of various types of data and analyses; however, the model does not deal with the analysis or visualization of AIS data [23]. Another method was developed to enhance responses to maritime emergencies by using the Electronic Chart Display and Information System. The system analyzes and displays optimized situation-dependent maneuvering plans for maritime traffic emergencies [24]. Further, hydrographic geospatial standards for marine data and information and their effective representation were suggested for communities, including marine science and maritime technology [25]. Additionally, a scenario for shipping industry was developed for stakeholders that employs analytics, stream processing, monitoring, alerting, and vessel route optimization over big data [26].

Some countries or cities have provided information for warning and guiding marine traffic by operating authorities. For example, according to the U.S. Coast Guard Navigation Center, the Coast Guard operates 12 vessel traffic centers (VTC) and 200 very-high-frequency (VHF) receiver sites located throughout the coastal areas of the United States [27]. In the case of Hong Kong, its Marine Department has 13 radars employed in the system to provide radar surveillance coverage of Hong Kong's navigable waters. The radar system is designed to automatically track a maximum of 10,000 targets at any one time [28]. The Swedish Maritime Administration has a network of land-based AIS base stations to receive AIS information from vessels and transmit safety-related information. The network uses AIS information to improve maritime safety information, search and rescue missions, and icebreaking operations [29]. Further, the Port of London Authority has a team of 44 VTS personnel that oversees its VTS area on a 24/7 basis, 365 days a year. A VTS supervisor with the delegated powers of the harbor master leads each VTS Center, and a team of VTS officers and shipping coordinators supports the VTS supervisor [30]. In South Korea, the Maritime Transportation Control Center of the Korea Cost Guard operates 20 VTS centers [31].

Many works have dealt with spatial and temporal approaches to analyzing AIS data on vessels. For instance, AIS data were utilized to measure the safety of a vessel's bridge and showed their potential impact on the safety of marine navigation. Furthermore, issues related to uncertainty that may be generated by different regulations to supervision for proper use, training, and management of AIS users were investigated [13]. AIS data were used for creating charts and analyzing individual vessel movements and general traffic patterns to quantify air pollutant emissions from the vessels [16]. Moreover, a prediction model of fuel oil consumption and a weather routing algorithm were used together to propose an optimal route for a vessel for fuel oil consumption efficiency using data collected from AIS, Sensors, Noon Reports, and Weather Service APIs [32]. An algorithm was developed for three-dimensional space–time density analysis and the visualization of AIS data collected from the Gulf of Finland [33]. An open-source toolbox was also developed and released that provides modules to support AIS data processing and visualizing traffic density analyzed on charts [34]. Additionally, a linear regression model was developed for AIS data to avoid collision by identifying the correlation of the closest point of approach, which is a crucial indicator for collision avoidance, to the vessel's size, speed, and course [14]. A methodology was developed to extract traffic routes, detect low-likelihood behaviors of vessels, and predict the routes of vessels from a large amount of AIS data. Moreover, the authors visualized the predicted destination of vessels and the probability of the observed tracks [15]. A comfortable navigational distance was estimated for vessels by analyzing the intensity plots generated from AIS data [12]. The risk of the collision of vessels was analyzed by using developed software and AIS data. The tool calculated the collision risk from traffic patterns in the data and visualized the results [18].

A user test on visualization tools is necessary to support user-oriented communication and decision making [35,36]. An agenda for empirical research on user interactive design for digital maps was developed [37]. Several works have conducted user studies on the use of maps and the cartographic visualization of spatial data. For instance, data from historical sites were visualized in a virtual reality application, and users tested the effectiveness and attractiveness of the application [38]. An online atlas contained usability research and adopted a user-centered design methodology for better decision making [39]. The usability of some mapping techniques for visualizing spatial accessibility was evaluated in terms of effectiveness, efficiency, graphical attractiveness, and user-perceived effectiveness [40].

However, most of these previous works have limitations regarding providing information to officers about multiple vessels moving concurrently, so that the officers can understand quickly because of their complex representations of information [21]. Therefore, in this research, we apply the REMO analysis method and visualization, the existing approaches in GIScience, to extensive marine traffic data, in order to improve the analysis and visualization of vessel movements and to help users better understand such information. The REMO analysis measures relative speed, change of speed, and the azimuth of moving objects to detect constancy, concurrence, dispersion, and so on. Studies have used REMO analysis to detect and measure constancy, concurrence, trendsetting, turn, opposition, and dispersion of moving objects such as herds, athletes, and dancers [19,41–44]. Further, several studies have extended the REMO approach. For example, the REMO approach has been used in studies to discover locational and temporal patterns of moving objects, such as caribou, soccer players and dancers [41,43–45]. Moreover, methods were developed to detect spatio-temporal patterns of flock, leadership, convergence, and encounter by following the REMO approach and using computational geometry. The researchers improved existing algorithms by decreasing running time bounds [42]. A framework was developed to detect the motion anomalies of vessels by conducting a statistical analysis of real-time AIS data [17]. In addition, a semantic recognition method was proposed to analyze ship motion patterns entering and leaving a port based on a probabilistic topic model. The analysis used a large amount of trajectory data in an unsupervised manner [46]. Moreover, a visual analytics tool was proposed that used spatial segmentation to identify vessels' anomalous trips and measure the degree of unusual vessel behavior [47]. Furthermore, the motion patterns of moving objects, such as dancers using REMO matrices and dynamic time warping, were visualized and analyzed. The motion attributes of speed, motion azimuth, vertical angle over time, and time-series analysis of the attributes were measured [45]. However, little research has investigated vessels' movements using the REMO approach. Therefore, using the REMO approach, we attempt to fill the gap in the literature by analyzing and visualizing the characteristics of multiple vessels' movements at a port based on the empirical AIS datasets of the vessels. Further, we develop a prototype software tool, "V-REMO", to interactively analyze the vessels' relative movements based on the AIS data and display the results on digital charts. For instance, the digital charts in V-REMO show changes in the azimuth, speed, and $\delta$-speed—or change of speed—of each vessel per time unit and location based on the user's manipulation. Finally, we conduct user evaluations of V-REMO to enhance the tool in the future.

## 2. Materials and Methods

### 2.1. Study Areas and Data

The study areas included the Port of Yeosu and nearby areas located in South Jeolla Province, South Korea, and the southwestern areas of the Korean Peninsula (Figure 1). The Port operates passenger terminals and cargo docks [48].

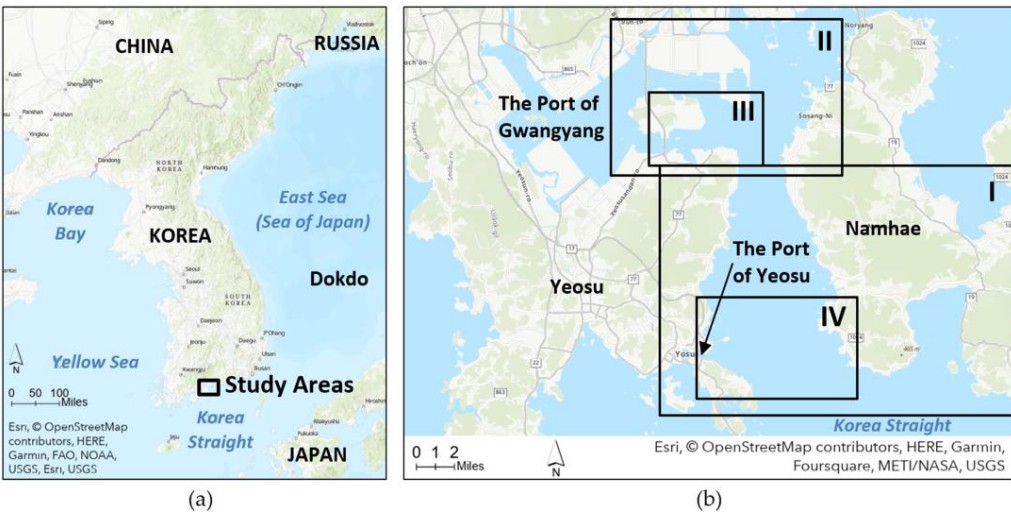

**Figure 1.** (**a**) Location of the study areas; (**b**) the study areas enlarged from (**a**): area I includes the Port of Yeosu and City of Namhae, area II includes Myo Island and Cities of Gwangyang, Namhae, and Yeosu, area III includes Myo Island and City of Yeosu, and area IV includes the Port of Yeosu and Cities of Yeosu and Namhae.

AIS prevents the collision of vessels by providing real-time navigation information such as the location, course, and velocity of each vessel. The International Maritime Organization (IMO) regulation requires AIS to be mounted on vessels of certain types [49] (p. 2). AIS can identify a vessel's existence, name, navigation direction, and velocity, even when the naked eye or marine radars cannot. Thus, maritime safety management activities, such as preventing vessel collisions, controlling broad areas, and searching and rescuing wrecked vessels, can be more efficient using AIS. In general, AIS data help to analyze vessels' navigation because they consist of multiple types of information on a vessel. Since AIS messages are encoded in binary format, they should be separated from other non-binary types of data in a database (Figure 2).

```
!AIVDM,1,1,,A,B6SWVSP002@p2o5FojNc<vkUoM06,0*4D,5,END
!AIVDM,1,1,,A,16SlPM0P00a2doBE;V8ev0E>2D3l,0*19,2,END
!AIVDM,1,1,,B,8>lEJr00Bj8C4kWNieId0000042P00000001HL0000000000000000h0000,2*0D,0,END
!AIVDM,1,1,,A,16SWbsPP00a8QuVAKcOjM?w>25AP,0*71,5,END
!AIVDM,1,1,,A,10aucQ0000a1vbPEY9a52HW@2@GV,0*6C,5,END
```

**Figure 2.** An example AIS message encoded in a binary format. Data from [50].

Generally, a VTS center operates multiple AIS-receiving stations that collect all AIS signals within the maritime territory, including the control areas. Therefore, a vessel's course characteristics per time can be studied by analyzing its AIS data accumulated over a certain period. AIS data primarily consist of the items in Table 1.

AIS receivers collect messages from vessels within signal reception areas per the order of the reception. Thus, AIS data are not distinguished per vessel. Additionally, AIS data are generated every few seconds or minutes according to the velocity of vessels. Therefore, the amount of AIS data received within a control area can be large when multiple vessels move concurrently. Table 2 shows the transmission frequency of AIS messages. AIS data have different time intervals since the data are not collected by a regular time unit. Therefore, the density of the data collection becomes higher when the velocity of a vessel increases.

**Table 1.** Items in AIS data.

| Items | Descriptions |
|---|---|
| Static information | - Identifies information about a vessel.<br>- Is transmitted every 6 min or per request.<br>- Includes information such as Maritime Mobile Service Identity, International Maritime Organization number, name of vessel, maritime call sign, beam and length, and type of vessel. |
| Dynamic information | - Is related to a vessel's movement.<br>- Transmits a vessel's frequency per velocity.<br>- Includes the location of the vessel, course over ground, speed over ground, heading, and angular velocity of the vessel's head. |
| Voyage related data | - Include vessel draft, types of cargo, destination, and estimated time of arrival. |

**Table 2.** Transmission frequency of AIS messages.

| Status of Vessel | Transmission Frequency |
|---|---|
| Ship at anchor | 3 min |
| Ship 0–14 knots | 12 s |
| Ship 0–14 knots and changing course | 4 s |
| Ship 14–23 knots | 6 s |
| Ship 14–23 knots and changing course | 2 s |
| Ship > 23 knots | 3 s |
| Ship > 23 knots and changing course | 2 s |

KRISO provided AIS data on the study areas. The data included 31,708,973 records of 212 vessels collected during 11–20 January 2012, in the Port of Yeosu. The AIS data were decoded for use by the REMO software tool.

*2.2. REMO Analysis*

In the study of moving objects in space and time, it is important to reveal information on movement attributes, such as direction and speed [33]. We adopted the REMO approach [19] to analyze the patterns of the relative movements of vessels, including the azimuth (or direction), speed, and $\delta$-speed of vessels. Figure 3 introduces example concepts of the REMO approach for vessels following Laube and Purves [43]. For example, in Figure 3a, vessel one moves towards the south with a constant motion azimuth of 135°. The movement of vessel one occurs during an interval from $t_1$ to $t_3$ (Figure 3b,c). The vessel's movement includes three discrete time steps of length $\delta$t and shows a constancy of motion azimuth (Figure 3d). The vessel's speed is measured based on the distance and duration between the two adjacent locations of the vessel's trajectory. The vessel's $\delta$-speed is calculated based on the differences between the speed of the two adjacent locations of the vessel.

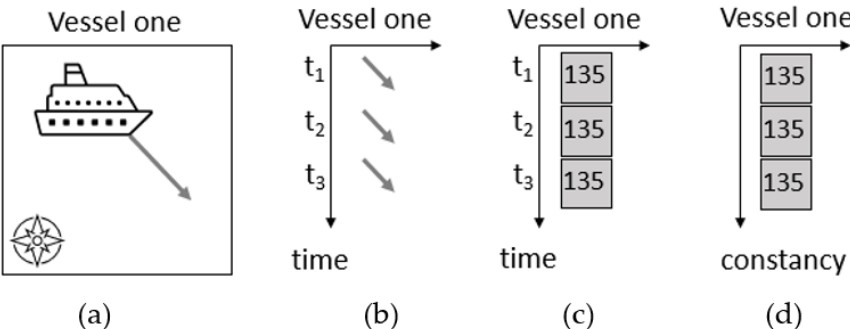

**Figure 3.** Example concepts of the REMO approach for vessels following [43]: (**a**) a vessel moves toward the south with a constant motion azimuth; (**b**) the constant motion azimuth occurs per time unit; (**c**) values of the motion azimuth, 135°, are coded per time unit; (**d**) the vessel's movement shows a constancy of motion azimuth.

The patterns of REMO can be graphically recognized in a "REMO matrix", a two-dimensional conceptual space consisting of a time axis and another axis of individual moving objects [19,44]. Based on the steps of the example in Figure 3, three REMO matrices of the vessels were generated from the database in this study. The REMO matrices consist of information about the motion azimuth, speed, and $\delta$-speed—or change of speed—of the vessels, respectively, following Laube and Imfeld [19]. Then, the motion azimuth of the vessel was measured based on the x and y coordinates between two adjacent point locations of the vessel from the dataset. The motion azimuth of the vessel was classified as the following: N (either 0~22.5° or 337.5~360°), NE (22.5~67.5°), E (67.5~112.5°), SE (112.5~157.5°), S (157.5~202.5°), SW (202.5~247.5°), W (247.5~292.5°), and NW (292.5~337.5°). Further, "○" was used for no movement in location. However, the "sorites paradox" in classifying angles may exist because of the crisp classification used in this study. For instance, two values, 22.5° and 22.51°, were almost identical. Nonetheless, they were classified into two classes, N and NE, respectively (Section 3.1). Finally, charts that showed empirical locations of the vessel's REMO were created to provide spatial patterns of the vessel's REMO (Section 3.2). The REMO matrices and the charts of the REMO analysis were constructed using ArcGIS Pro (10.8) [51].

In the next section, we introduce V-REMO, a software tool that supports the REMO analysis on the AIS data.

### 2.3. Development of Software, V-REMO

This section describes the development process of V-REMO, a prototype software tool for marine traffic. V-REMO was developed by the project team at KRISO to analyze multiple vessels' real-time movements from the decoded AIS data. The tool provides information for safe navigation based on the Microsoft NET framework. The tool aims to help the decision-making process of vessel traffic control officers and reduce their workload. V-REMO consists of a menu pane and a chart view pane to visualize analysis results on digital charts. The digital chart is displayed as the background on the chart view pane because such charts are familiar to navigation officers and controllers, and they know how to use them. The digital chart provides data on topics such as the fairway, aid to navigation, and the water depth, so that information affecting a vessel's navigation can be grasped. The digital chart is displayed as raster data on the user interface (UI) of V-REMO to shorten the data loading time.

A list of the vessels' Maritime Mobile Service Identity (MMSIs) from the database is displayed on the left-hand side of the UI. When the user selects a vessel, the vessel's trajectory is displayed in a different color from other vessels' trajectories on the digital chart. The user can display changes in the selected vessel's trajectory per time by using a time slider at the bottom of the UI. The user can select SOG, COG, $\delta$-SOC (or change of SOC), and $\delta$-COG (or change of COG) in the UI to visualize the analysis results. SOG

and $\delta$-SOG refer to information about the vessel's velocity, and COG and $\delta$-COG refer to information about the vessel's course. Usually, AIS data include information on the heading of a vessel and its rate of turning (ROT). However, AIS data were not used to analyze the vessels' trajectories in this study because the ROT values represent the intention of a vessel's navigation rather than the results of its navigation. Finally, the user can select one of the analysis results from V-REMO and adjust its graphic symbology, such as the outline and transparency, to enhance the display. Figure 4 shows the process of the REMO analysis of a vessel's movement (Figure 4a) and the visualization of the results (Figure 4b) in V-REMO.

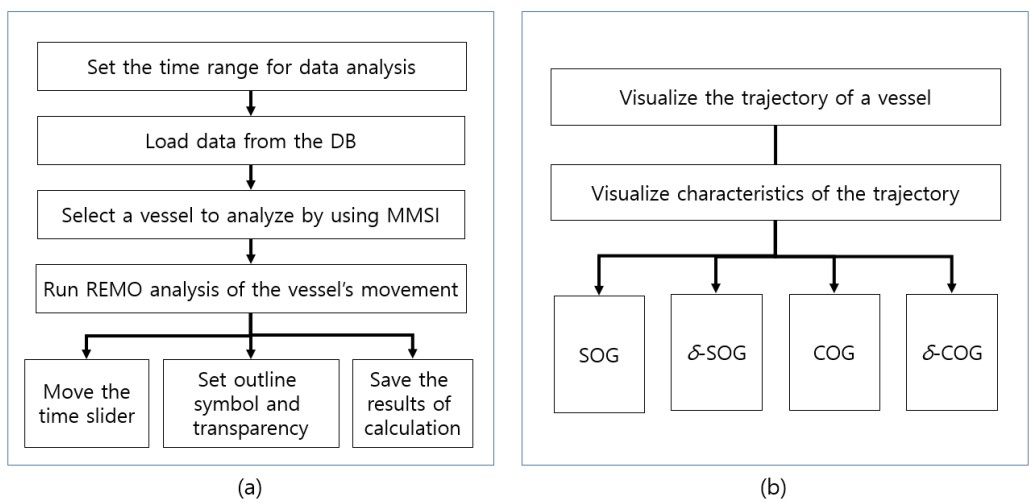

(a)                                                         (b)

**Figure 4.** Functions of V-REMO ((**a**): flow of user's manipulation; (**b**): information view).

Figure 5 shows the flow of the development and assessment of V-REMO in this study. It consists of data preparation, the development of V-REMO, the analysis of results, and tool assessment. In the next section, we describe the assessment of V-REMO in detail.

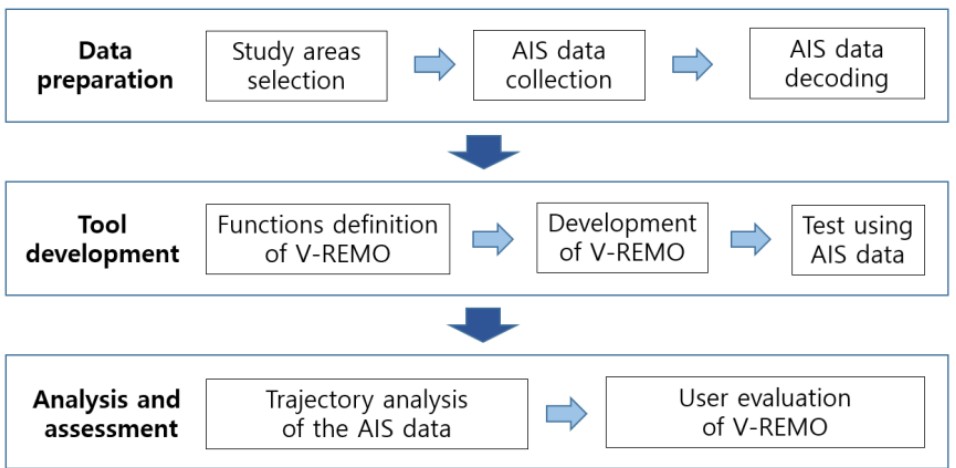

**Figure 5.** Structure of the study.

### 2.4. User Evaluation of V-REMO

The usability of an application for maritime transportation should be tested in close collaboration with the intended users, including the analysts at VTS stations and maritime authorities [33]. We conducted user evaluations of V-REMO to test its usability. We examined the prototype of V-REMO by conducting in-depth interviews with stakeholders in South Korea to determine whether V-REMO's UI supports data analysis and trajectory recognition activities, and to identify its existing limitations for its further development.

The participants in the interviews were six professionals in the maritime domain, including researchers and navigation officers. The participants were recruited because they had diverse specializations, including marine safety, marine transportation, maritime standards, product specification, e-navigation, and ergonomics. The participants' ages varied from those in their 20s to those in their 40s. Their work experience in the industry varied from 1 to 18 years. None of them was involved in any processes of the development of V-REMO. All subjects gave their informed consent for inclusion before participating in the study. The study was conducted in accordance with the Declaration of Helsinki, and the protocol was approved by the Institutional Review Board of California State University, Long Beach (Project number: 1146800-2, Reference number: 18-149).

The user evaluation consisted of practical demonstrations of V-REMO and in-depth interviews. First, each participant was informed about the purpose of the user evaluation and the concept of the REMO analysis of the vessels' trajectories. Then, examples of demonstrations using V-REMO were provided, and each participant was asked to answer questions about their own experience of using V-REMO. Finally, the in-depth interviews comprised 26 questionnaires regarding demographic information on the participants, the general usability of V-REMO, the data visualization, and the readability of information in the UI. For example, the participants were asked to evaluate the UI's graphic design and the usability of the menus and buttons, the chart design of the REMO analysis results on the digital chart, and the readability of the real-time traffic of vessels in the UI (figures are provided in Section 3). The questionnaires used in the user evaluation of this study are provided in Supplementary File S1.

## 3. Results

In this section, we provide results of the REMO analysis on AIS data. Then, we show the developed V-REMO and describe the results of the user evaluations.

### 3.1. REMO Analysis

First, we introduce the hypothetical results of the REMO analysis in Section 3.1. The AIS data used in this section were produced arbitrarily to provide an example of the REMO analysis. Then, we provide the results of the REMO analysis using the empirical AIS data collected in the study areas in Section 3.2.

Figure 6 presents the initial results of the REMO analysis regarding the azimuth (Figure 6a), speed (Figure 6b), and $\delta$-speed (Figure 6c) of vessels 1, 2, 3, and 4 in REMO matrices. The four rows in each REMO matrix in Figure 6 visualize the relative movements of the corresponding vessels. Since the four vessels started their navigation at different times, the starting time of their navigation and the lengths of their data in the REMO matrices are not identical. Figure 6d–f show a part of the REMO matrices enlarged from Figure 6a–c, respectively. In general, the motion azimuth of the vessels shows a variety of their moving directions ("divergence" in [19]) during the navigation (Figure 6a). Some vessels changed their directions (the color hue of a row changes in Figure 6d ("turn" in [19])), while others did not (the same color hue continues to appear in the row in Figure 6d ("independence" in [19])). Blank cells represent no data, since some vessels often had no difference of azimuth. Sometimes, vessels' movements that are not clear in the azimuth matrix (Figure 6d) become more explicit in the speed matrix (Figure 6e) or the $\delta$-speed (Figure 6f). For instance, area I in Figure 6d does not show any changes in the azimuth of the vessels. However, area II in Figure 6e and area III in Figure 6f show variations in the speed and $\delta$-speed for the same data as Figure 6d, respectively.

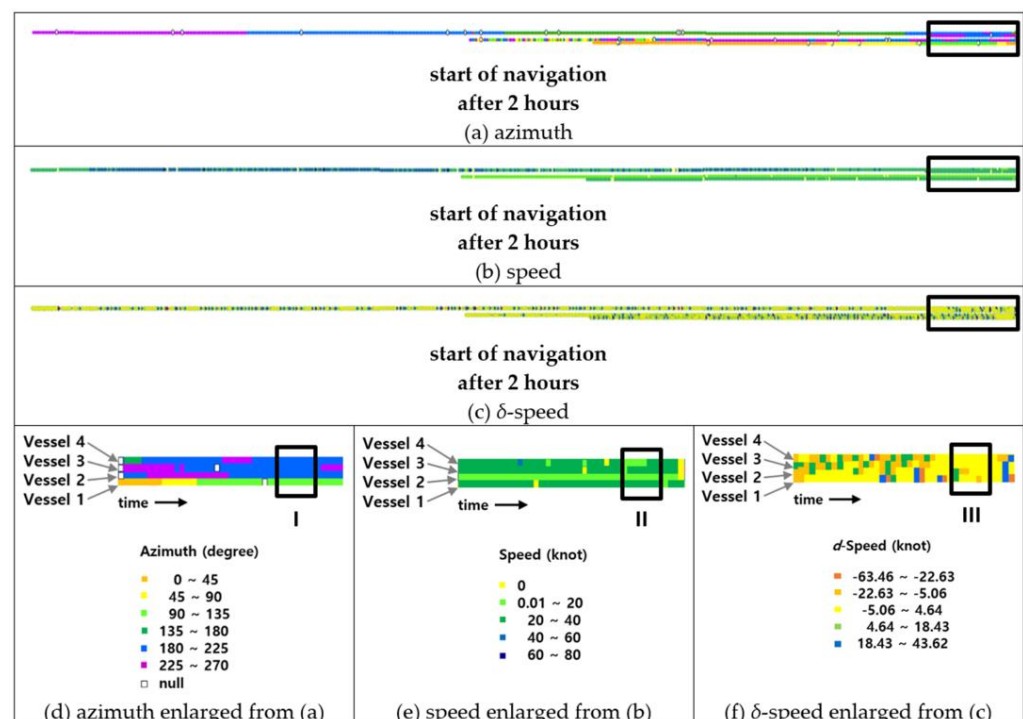

**Figure 6.** Example REMO matrices of the four vessels during two hours of navigation: (**a**) REMO matrix of azimuth, (**b**) REMO matrix of speed, (**c**) REMO matrix of δ-speed, (**d**) part of (**a**) enlarged, (**e**) part of (**b**) enlarged, (**f**) part of (**c**) enlarged.

Figure 7 spatially visualizes the REMO analysis results of the four vessels in terms of azimuth (a), speed (b), and δ-speed (c) during the whole navigation. Figure 7a shows the relative azimuth of each vessel at its location. In Figure 7a, yellow-green, green, or blue colors indicate a substantial change in azimuth, and yellow, orange, or purple colors indicate a small change in azimuth. The color hues in Figure 7a change where the azimuth of each vessel changes during their navigation. Figure 7b shows the relative speed of each vessel at its location. The relative speed of the data clearly shows that the speed of the vessel "MMSI: 636091308" changed, although its azimuth was maintained while navigating in area I in Figure 7b. Finally, Figure 7c shows the relative δ-speed of each vessel at its location. The relative δ-speed, or change of the speed, of the data visualizes an increase (in green or blue color hue) or a decrease (in red or orange color hue) in the speed of each vessel while navigating (Figure 7c).

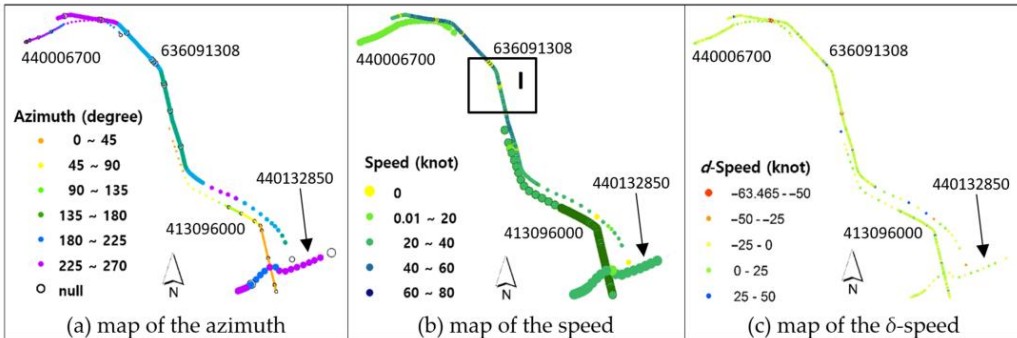

**Figure 7.** Charts showing REMO analysis of the four vessels during the navigation in terms of azimuth (**a**), speed (**b**), and δ-speed (**c**).

*3.2. Application: Empirical REMO Analysis Using V-REMO Software*

This section provides the results of the REMO analysis on empirical AIS data collected in the study areas as an application. The REMO analysis was conducted by using the V-REMO software developed in this study. The UI of V-REMO primarily consists of a data processing section and a chart visualization section. First, a user can load the trajectory database and specify the time range to query (A of Figure 8). Once the trajectory data are loaded, a list of vessels is displayed, including each vessel's MMSI number (B of Figure 8). Then, the user can select a vessel from the list and move the time slider to visualize the vessel as a symbol on the electronic navigational chart along the timeline (G of Figure 8). Thus, the user can see the trajectories of each vessel and select a vessel to analyze and visualize the vessel on the electronic navigational chart (C of Figure 8). The user can preprocess the trajectory data on the selected vessel and change the size and color of the vessel's symbol by using the "Calc" button. Additionally, the user can adjust the transparency of the electronic navigational chart to enhance the visibility of information. Figure 8 shows the UI of the V-REMO software.

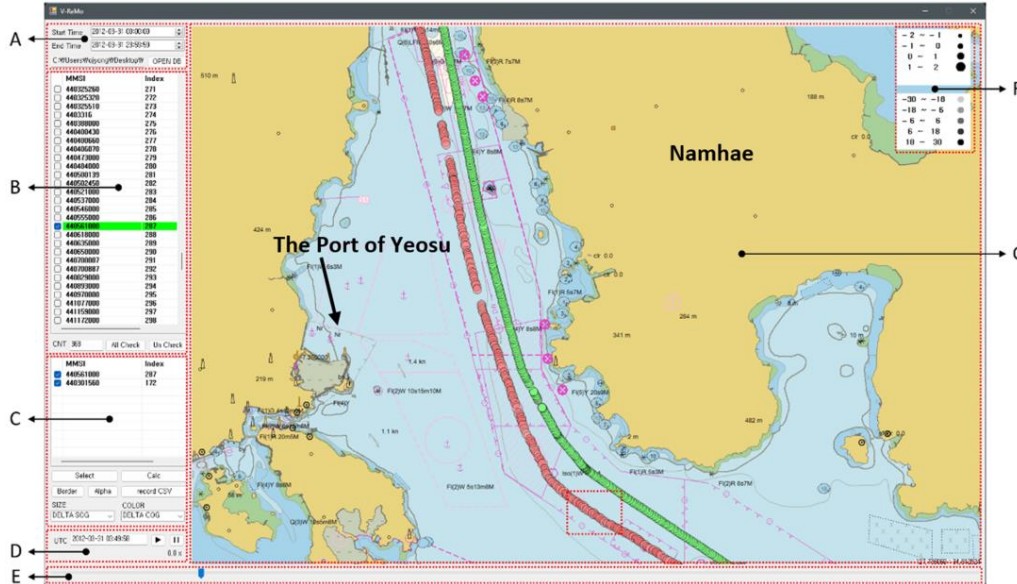

**Figure 8.** The UI of V-REMO showing area I in Figure 1b: (**A**) the vessels' empirical AIS data are loaded from a trajectory database, and the time range for a query is set; (**B**) vessels in the database are selected; (**C**) vessels to be analyzed are selected, and visualization attributes are set; (**D**) the playback of trajectory data is controlled; (**E**) the slider bar is controlled; (**F**) the symbols of the vessels are displayed in the legend; and (**G**) the electronic navigational chart is shown in the chart view.

One of the unique characteristics of the V-REMO software is it can efficiently visualize information for the REMO analysis of the vessel's trajectory data. AIS collects the trajectory data. The data consist of MMSI numbers, the location information (longitude and latitude) of the vessel, the SOG and COG of the vessel, time, and so on. Among these, the values of SOG and COG are closely related to the vessel's movement characteristics. The V-REMO software measures the vessel's movement over time and visualizes the results using symbols on the digital chart. A vessel's symbols are located on the vessel's trajectory and are displayed in different sizes and colors depending on SOG, COG, $\delta$-SOG, and $\delta$-COG values. Figure 9 provides an example of SOG changing along with the vessel's movement and the amount of change.

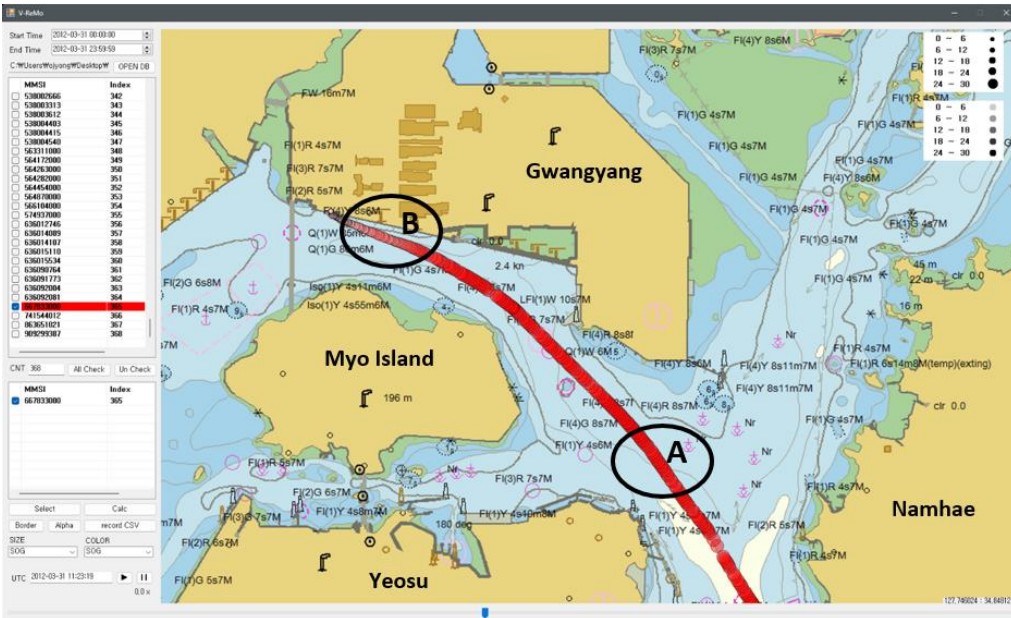

(A) The UI of V-REMO showing the area II in Figure 1(b)

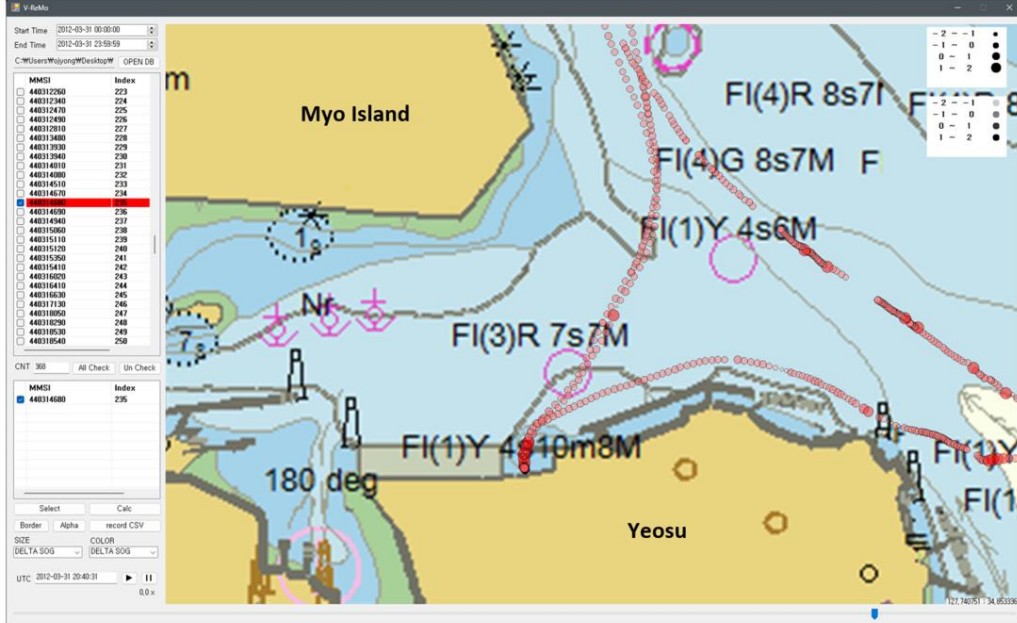

(B) The UI of V-REMO showing the area III in Figure 1(b)

**Figure 9.** Example of vessel trajectory visualization: (**A**) a vessel's SOG while approaching the pier at Gwangyang Bay; (**B**) a vessel's symbols showing the change of SOG during the movement.

Figure 9A visualizes information on the trajectory of a vessel during its pier approach: the size and color of the vessel's symbol change following its speed change. For example, the symbol becomes more prominent and has higher color saturation when the vessel's speed increases (Area A in Figure 9A). The user can see the vessel decreasing its speed as it approaches the pier (Area B in Figure 9A). Figure 9B displays the trajectory of a vessel moving to a pier within a port: the size and the color of the vessel's symbol change following its speed change in five classes. The user can easily recognize the location where the vessel's speed suddenly changed and the time when it changed.

Additionally, the user can replay the vessel's movements in the trajectory data through time using the V-REMO UI. The user can control the playback time of the trajectory data by moving the slider bar beneath the UI and by visualizing the trajectory data on the chart. By

doing so, the user can analyze the vessel's trajectory per time and compare it with that of other vessels.

Figure 10A,B displays the trajectories of the two vessels MMSI: 440144600 (in green) and MMSI: 440132190 (in red) at $t_1$ and $t_2$, respectively. The two vessels may appear to meet in Figure 10B; however, their trajectories per time in Figure 10A indicate that they did not meet.

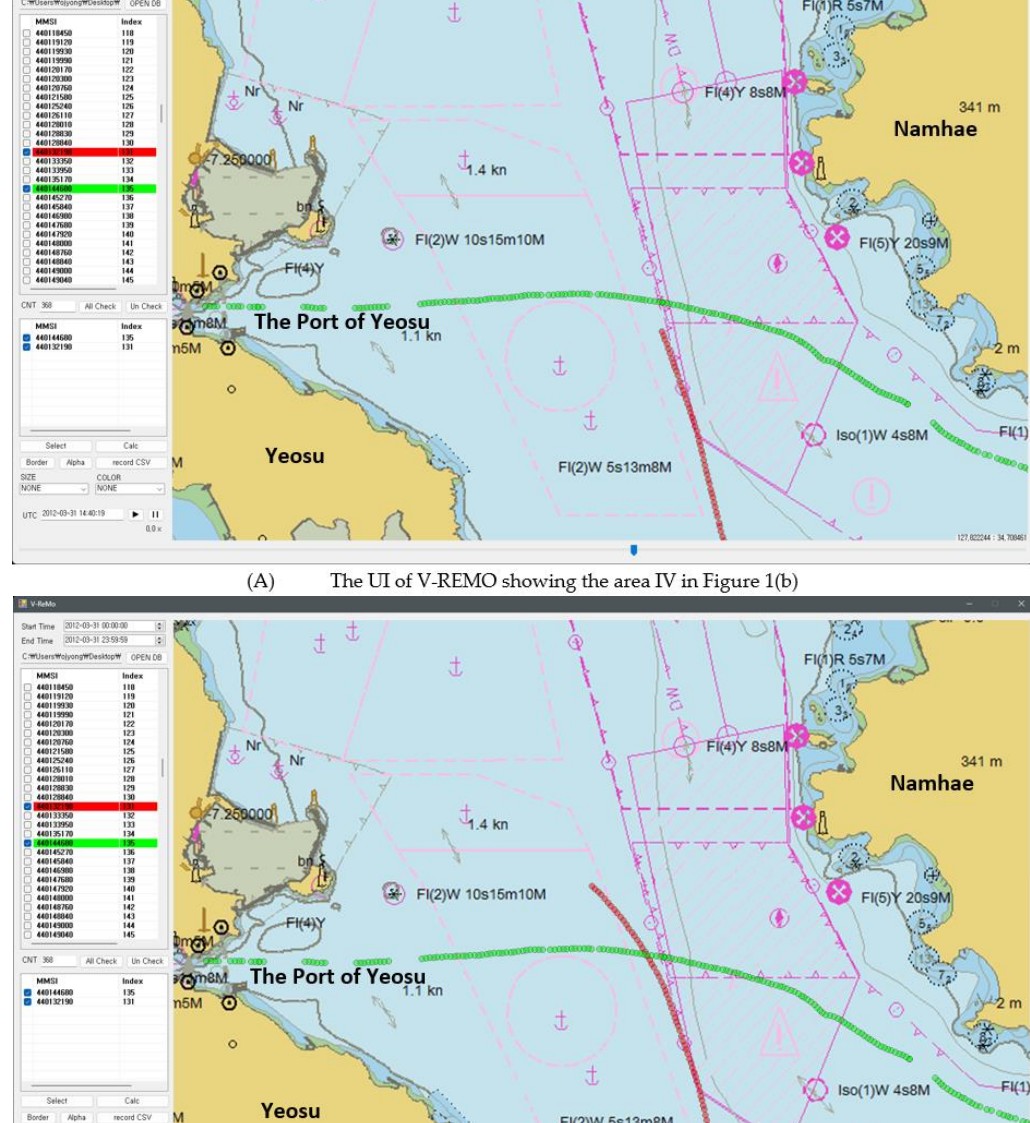

(A)     The UI of V-REMO showing the area IV in Figure 1(b)

(B) The UI of V-REMO showing the area IV in Figure 1(b)

**Figure 10.** Example visualization of the vessel's trajectories: (**A**) trajectories of two vessels, MMSI: 440144600 (in green) and MMSI: 440132190 (in red) at $t_1$ = 14:40:19; (**B**) the same trajectories at $t_2$ = 14:44:36.

### 3.3. User Evaluations of V-REMO

The usability of the V-REMO software was assessed in the user evaluations. The in-depth interviews consisted of a few questionnaires on the usability of V-REMO. First, regarding the general usability of the UI, the interviewees responded that the V-REMO

software tool was easy to manipulate and the electronic navigational chart in the UI was user-friendly. However, the interviewees also pointed out that the querying functions were too limited to represent a vessel's trajectories. For instance, one user stated that querying vessels' trajectories was performed quickly using only time ranges in the current V-REMO. However, finding a single vessel to analyze its trajectories from the data was challenging. This was because the user misunderstood the characteristics of V-REMO. For example, the user was trying to query a specific vessel's trajectory throughout the timeline from the dataset. However, the UI was designed to compare multiple vessels' movements during a specific time range rather than a vessel's movements. This suggests that users of V-REMO may need to have good knowledge of REMO analysis and V-REMO before using the software tool. Further, the interviewees mentioned that the menus in the UI were not easy to find and too many menus were shown. The interviewees also pointed out that querying specific information for individual vessels was not possible.

Second, regarding the issues in visualizing analysis results in the V-REMO UI, the interviewees responded that the visualization of the tool is convenient for understanding the trajectories of a vessel. However, most interviewees could not recognize the analysis results easily. Namely, they found it difficult to differentiate the colors and sizes of the multiple vessels' symbols. They specified that the circular symbol representing the vessels looked identical to another type of symbol in the electronic navigational chart. Therefore, it is necessary to examine the standard of the electronic navigational chart for symbolization.

Finally, regarding the readability of information on the real-time traffic of the vessels in the UI, some interviewees responded that they could recognize moving vessels because their trajectories and attribute information were displayed together on charts. However, most of them pointed out that it was challenging to understand the vessel's traffic in the UI when multiple vessels appeared or when the vessels' trajectories were cluttered on the electronic navigational chart.

The interviewees suggested the following for improving V-REMO. First, they recommended providing more options for querying the trajectory of a vessel in the UI. For example, the user should be able to query the trajectory data on a vessel by using the name, MMSI, type, and size. Additionally, the user should be able to view specific information about the vessel, including specifications and nationality, by selecting its symbol on the digital chart. The interviewees also recommended adding a function to display only a vessel's trajectories on the digital chart. Often, a single vessel may have multiple trajectories to display for a particular time range that the user defines. Thus, the digital chart of V-REMO can quickly become cluttered when multiple vessels are selected and their trajectory histories are displayed for the same time range.

Finally, the interviewees provided multiple suggestions for visualizing results in V-REMO. For instance, symbols of vessels should be able to represent the speed and course of each vessel. Additionally, the size and color of the symbol showing the analysis results should be clearly distinguished from other symbols. In particular, users can confuse circular symbols with other symbols in the digital chart that are not appropriate to represent the speed and direction of navigation. More studies are necessary to symbolize vessels following IMO's Guidelines for the Presentation of Navigation-Related Symbols [49]. The interviewees also proposed showing the navigation data on all vessels using a regular time unit. However, this suggestion does not apply to the V-REMO UI because periodic intervals of the transmission of navigation data can be an essential factor for marine transportation analysis.

The user evaluation examined the UI design and the software functionality of V-REMO. In terms of the UI design, the users wanted to see changes in the size, position, colors, and transparency of the trajectory symbols in the digital chart for better readability and easier manipulation. The users also indicated a need for multiple color schemes for the spatial data and more landmark information about the study area in the chart view. The responses from the user evaluation participants of this study are provided in Supplementary File S2. Additionally, the user feedback contributed to discovering existing limitations of V-REMO

that needed to be rectified, including supporting users in understanding how it works and designing some functions and visual displays in the UI.

## 4. Discussion

This study demonstrated how the REMO approach and visualization could analyze multiple vessels' movements from empirical AIS data. The significance of this research is that it expands the range of GIScience to the domains of ocean engineering and maritime transportation. Our research introduced a new application of the REMO approach to enhance the work environment of maritime control officers. In South Korea, maritime control officers at all six selected VTS centers out of eighteen centers have a high workload that may lead to human errors in recognizing information about vessels' status and making decisions in real-time [5]. Therefore, it is necessary to develop a system to decrease officers' workload so that they can analyze complex maritime information.

Despite the novel approach of this study, there exist a few limitations. We discuss some major issues in three categories as examples. First, issues of the data: This study analyzed and visualized the AIS data in only two dimensions. Analysis and visualization of the data in three or higher dimensions in V-REMO—i.e., by including time as an additional dimension—might provide more insight about the data than the current study. Moreover, in general, AIS can provide data in a high resolution (i.e., record per second). The high resolution often makes AIS data useful for maritime transportation analysis [10]. However, other types of data related to maritime transportation, such as climate (i.e., 30 m), topography (i.e., 30 m), and weather (i.e., 10 km) have lower resolution than AIS data [52–54]. For this reason, the current study did not address the data of the climate, topography, and weather of the study areas. In addition, the results of this study might not be useful in understanding the movement of small vessels if they are not equipped with AIS [55]. Therefore, additional data types—for instance, very-high-frequency (VHF) radio data—should be considered in the analysis for more inclusive research.

Second, issues of V-REMO: The tool developed in this study did not deal with real-time AIS data due to the limitation of the pilot version of the software. Data collection, analysis, visualization, and traffic control for vessels should be in real-time for the better prevention of maritime transportation accidents than analyzing non-real-time data. Therefore, the functionality of V-REMO needs to be enhanced to deal with real-time data through interactive analysis and visualization.

Third, applying the REMO approach: The REMO approach in this study could be challenging for stakeholders in the maritime domain to understand if they do not have knowledge of the approach. The user evaluations of this study revealed that users should first understand the REMO approach before they use V-REMO. Moreover, users' needs for the analysis of AIS data should be investigated before a tool is developed to address their needs in the UI of the tool.

## 5. Conclusions

In this article, we present a new application of the REMO analysis and visualization on moving objects in space and time. This is a new approach in the research of the maritime transportation domain to deal with the movement of vessels. Further, it should be considered how the approach of this study compares with a similar approach for the analysis of maritime traffic data. To demonstrate the utility of the new application, we tested the REMO approach on an empirical dataset of moving objects representing maritime traffic in the Port of Yeosu. The results attempted to answer the research questions of this study as follows. First, an ideal maritime traffic control system should implement specific characteristics, including knowledge of the relative motion of vessels in the same areas, to support control officers in making decisions. Second, it is crucial to effectively visualize multiple vessels' navigation information to support the maritime traffic control system effectively. Specifically, the user evaluation results in this study demonstrated the importance of human factors in software development for maritime traffic control. For

instance, users' understanding of the REMO analysis of AIS data can be largely affected by the symbolization in the software UI. Finally, the study provided new insights into the collaboration between geography and ocean engineering disciplines dealing with maritime traffic safety.

Future extensions of this study will include the development of a decision-making system based on real-time maritime traffic data by embedding artificial intelligence (AI) and the V-REMO functionality to reduce the labor of maritime traffic controllers. Therefore, it is necessary to develop an ontology-based context-aware system for maritime traffic accidents to embed AI in the decision-making system. In addition, the UI design of the V-REMO will be enhanced by applying the user evaluation results of this study. Finally, the variables of the closest point of approach (CPA)/time to closest point of approach (TCPA) between two adjacent vessels will be included in the dataset to measure the probability of collision between them. The CPA/TCPA data in the V-REMO will be able to support maritime transportation safety.

**Supplementary Materials:** The following supporting information can be downloaded at: https://www.mdpi.com/article/10.3390/ijgi12030115/s1, Supplementary File S1: the questionnaires used in the user evaluation of this study translated in English; Supplementary File S2: the responses from the user evaluation participants of this study translated in English.

**Author Contributions:** Hyowon Ban and Hye-jin Kim contributed to the conceptualization; methodology; software; validation; formal analysis; investigation; resources; data curation; writing—original draft preparation, review and editing; visualization; supervision; and project administration of the study. Hye-jin Kim contributed to the funding acquisition. All authors have read and agreed to the published version of the manuscript.

**Funding:** This research was supported by a grant from the Korean National Research and Development Project, "Development of Technology of VTS Big Data and Safety Information System," funded by the Korea Coast Guard, Republic of Korea (1535000332-201904962).

**Data Availability Statement:** Restrictions apply to the availability of these data. Data were obtained from KRISO. Please contact http://www.kriso.re.kr (accessed on 7 March 2023) for inquiries.

**Acknowledgments:** The authors acknowledge the contribution made by the project team members at KRISO, who provided technical support for the development of V-REMO.

**Conflicts of Interest:** The authors declare no conflict of interest. The funders had no role in the design of the study; in the collection, analyses, or interpretation of data; in the writing of the manuscript; or in the decision to publish the results.

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
