# Peer review of "Analysis and Visualization of Vessels’ RElative MOtion (REMO)"

_ijgi, doi:10.3390/ijgi12030115_

Round 1

Reviewer 1 Report

See the attached PDF document.

Reviewer 2 Report

This is an interesting applied research paper  that demonstrates the main pillars of an application regarding traffic control management in the maritime sector. We believe that with further research and evaluation, through user requirements elicitation, this software can be extended into a stand-alone commercialized application including collision avoidance modules, voyage planning as well as port traffic optimization. Our only concern regards the related work presentation, as from our point view some works are worth mentioning and take into consideration. A list of these works can be found in the pdf attached.

Reviewer 3 Report

The paper presents a tool and a user study analysis of the effectiveness of REMO analysis in a maritime context. 

The novelty of this work can be found in the user study, but this should be reinforced in the introduction to clarify this argument. 

Regarding the literature review, I believe it is a bit outdated, mainly regarding user studies for risk-based tools in the maritime domain. For example, a quick search on google scholar shows some papers that also conduct user studies for risk tools in this domain or motion patterns analysis, such as: 

Abreu, Fernando HO, et al. "A trajectory scoring tool for local anomaly detection in maritime traffic using visual analytics." ISPRS International Journal of Geo-Information 10.6 (2021): 412.

Li, Gaocai, et al. "Semantic Recognition of Ship Motion Patterns Entering and Leaving Port Based on Topic Model." Journal of Marine Science and Engineering 10.12 (2022): 2012.

The tool presentation is well done, but the interview questionnaire and answers are missing. There are discussions about the results but it is not formally introduced and discussed. For user studies research, this is essential. 

Round 2

Reviewer 3 Report

All my comments were addressed.

Author Response

The authors very much appreciate the comments by Reviewer 3.